# Functional Analysis of Autophagy-Related Gene *ATG12* in Potato Dry Rot Fungus *Fusarium oxysporum*

**DOI:** 10.3390/ijms22094932

**Published:** 2021-05-06

**Authors:** A. Rehman Khalid, Shumin Zhang, Xiumei Luo, Hamayun Shaheen, Afshan Majeed, Mehdi Maqbool, Noosheen Zahid, Junaid Rahim, Maozhi Ren, Dan Qiu

**Affiliations:** 1School of Life Sciences, Chongqing University, Chongqing 401331, China; luox@cqu.edu.cn (X.L.); renm@cqu.edu.cn (M.R.); 2Department of Plant Pathology, University of Poonch Rawalakot, Azad Jammu and Kashmir 12350, Pakistan; 3School of Preclinical Medicine, North Sichuan Medical College, Nanchong 637000, China; zhangshumin@cqu.edu.cn; 4Department of Botany, University of Azad Jammu and Kashmir, Muzaffarabad 13100, Pakistan; hamayun.shaheen@uajk.edu.pk; 5Department of Soil and Environmental Sciences, University of Poonch Rawalakot, Azad Jammu and Kashmir 12350, Pakistan; afshanmkhan123_rkt@yahoo.com; 6Department of Horticulture, University of Poonch Rawalakot, Azad Jammu and Kashmir 12350, Pakistan; mehdimaqbool@upr.edu.pk (M.M.); noosheen.zahid@upr.edu.pk (N.Z.); 7Department of Entomology, University of Poonch Rawalakot, Azad Jammu and Kashmir 12350, Pakistan; junaid.rahim@upr.edu.pk

**Keywords:** autophagy, *FoATG12*, *F. oxysporum*, potato dry rot, virulence, filamentous fungi

## Abstract

Autophagy is an intracellular process in all eukaryotes which is responsible for the degradation of cytoplasmic constituents, recycling of organelles, and recycling of proteins. It is an important cellular process responsible for the effective virulence of several pathogenic plant fungal strains, having critical impacts on important crop plants including potatoes. However, the detailed physiological mechanisms of autophagy involved in the infection biology of soil-borne pathogens in the potato crop needs to be investigated further. In this study, the autophagy-related gene, *FoATG12*, in potato dry rot fungus *Fusarium oxysporum* was investigated by means of target gene replacement and overexpression. The deletion mutant ∆*FoATG12* showed reduction in conidial formation and exhibited impaired aerial hyphae. The *FoATG12* affected the expression of genes involved in pathogenicity and vegetative growth, as well as on morphology features of the colony under stressors. It was found that the disease symptoms were delayed upon being inoculated by the deletion mutant of *FoATG12* compared to the wild-type (WT) and overexpression (OE), while the deletion mutant showed the disease symptoms on tomato plants. The results confirmed the significant role of the autophagy-related *ATG12* gene in the production of aerial hyphae and the effective virulence of *F. oxysporum* in the potato crop. The current findings provid an enhanced gene-level understanding of the autophagy-related virulence of *F. oxysporum*, which could be helpful in pathogen control research and could have vital impacts on the potato crop.

## 1. Introduction

Autophagy is a conserved intracellular degradation and recycling process in eukaryotes which helps to overcome nutrient deficiency and allows cells to survive in starvation conditions. The process involves the recycling of degraded products, the synthesis of the macromolecules in vacuole or lysosome, and the formation of autophagosomes an important component of autophagy [1]. Recently, about 30 genes have been identified that control autophagy-related cellular mechanism in plants. Among these genes, *ATG8* and *ATG12* act as the principle components for formation of autophagosomes during nonselective macroautophagy [2]. The ubiquitin protein *ATG12* is activated by *ATG7* and *ATG10*, which forms a multimeric complex of *ATG12*-*ATG5*-*ATG16* that facilitates the lipidation of *ATG8* during the development of autophagosomes [3]. In yeast, the deletion mutant of *ATG8* and *ATG12* showed imperfect autophagy [4].

Autophagy is a key mechanism in the plant pathogenic fungal strains and plays significant role in cellular differentiation as well as pathogenicity [5,6,7,8,9]. Experiments on autophagy-related genes in rice blast fungus *Magnaporthe oryzae* proved that the absence of these genes in the mutants leads to failure in causing the disease. It was revealed that autophagy hinders the appressoria formation, which is the main disease-causing structure of the *M. oryzae* [10,11,12]. Autophagy has also been reported to decrease the virulence of corn smut fungus *Ustilago maydis* by stopping the formation of appressorium [9]. Studies have also identified several autophagy-related genes including *VdPKAC1*, *VdSNF1*, *VGB*, *VGB*, and *VMK1*, which are responsible for the pathogenicity in *Verticillium dahlia* [13,14,15,16].

*F. oxysporum* is the common fungal pathogen of the potato crop which infects tubers and roots [17,18]. The fungus attacks the crop at multiple stages causing wilt during the planting stage, whereas it causes dry rot at post-harvest storage stage, which has severe impacts on the crop’s nutritional as well as economic value [19,20]. The estimated annual yield losses in potato crops attributed to the *F. oxysporum* range between 6% and 25% in the field, whereas post-harvest storage losses have been recorded up to 60% [21,22,23]. *F. oxysporum* is a seed/soil-borne pathogen that mainly colonizes the plant vascular system and decomposes the tissues [24,25]. Potato dry rot is characterized by shriveling and shrinking of the root and tuber tissues turning the internal parts into black-brown color, along with having lesions on the surface [24,26,27]. Normally, the rot infection is induced into plant tissues by the pathogen and penetrated through wounds. The fungus develops streaks risen to the ground level turning the infected tissues dark red, whereas the infected leaves get pale yellow and detach from the plant with the passage of time [17]. It has been proven that autophagy controls the division and number of nuclei in the non-septate hyphae during the vegetative growth of the infected crop, vital for the virulence of *F. oxysporum* [28].

*ATG12* is an important gene which controls cellular mechanisms and virulence in several plant-infecting fungal strains. Although a number of autophagy-related genes have been investigated in the *F. oxysporum*, the role of *ATG12* in cellular mechanisms and determining the pathogenicity of the fungus still needs to be evaluated, indicating a considerable knowledge gap. The current research was designed to investigate the role of this important autophagy-related gene, i.e., *ATG12*, in the developmental processes and pathogenicity of the *F. oxysporum*. The *ATG12* overexpressed and the gene deletion mutants of *F. oxysporum* were generated by employing a target gene replacement technique. The specific objectives of the study were to investigate the role of these mutants on the development of fungal structures, including aerial hyphae and conidia, and to analyze the pathogenicity and survival of the fungus against stressors.

## 2. Results

### 2.1. Gene Replacement of OEFoATG12 and ∆ATG12

Deletion mutant genes of *F. oxysporum* strains containing GFP fusion proteins were created by performing replacement strategy in order to reveal the *FoATG12* target gene functions (Appendix A). PCR was used to analyze the hygromycin-resistant (HygR) transformants in which the expected shift in the FoATG12 transformant indicated the successful replacement of the *FoATG12* gene (Appendix A). PCR further confirmed the putative deletion mutants (Figure 1). Co-transformation of the *FoATG12* gene was carried out fused with the protein GFP to achieve the overexpression of the *FoATG12* gene, which was further validated by the PCR results in which the expected overexpression was shown by the transformants. 

### 2.2. Impaired Fungal Development Exhibited in the ∆ATG12 Mutants

The phenotypic characteristics of the fungal colonies and the conidia formation in the deletion mutants (∆*FoATG12*) along with the overexpression (*OEFoATG12*) transformants were investigated to analyze the impact of the *ATG12* gene on the growth and development of *F. oxysporum*. The colony formation of ∆*FoATG12* was different compared to *OEFoATG12* and wild-type (WT). After 7 days of inoculation, the growth rates of all the strains on PDA media were different from each other. However, ∆*FoATG12* produced less dense mycelium and slow radial growth compared to the *OEFoATG12* and WT (Figure 2A,B). Moreover, the conidial germination was measured after 7 days of inoculation, and the numbers of the recovered microconidia from ∆*FoATG12* were reduced significantly compared to WT and *OEFoATG12* (Figure 2C).

The potato dextrose broth (PDB) microscopic examination showed that ∆*FoATG12* inhibited hyphal growth compared to *OEFoATG12* and WT (Figure 3A), while extension in hyphae was observed after 24 h. The measurement of hyphal extension showed that the hyphae of ∆*FoATG12* were not significantly (*p* > 0.05) shorter than the *OEFoATG12* and WT transformants (Figure 3B). Taken together, these finding suggest that *F. oxysporum ATG12* contributes to conidial formation and hyphal formation. 

### 2.3. Response against Stress

In response to pathogen invasion, plants change the internal environment of the vascular system and apoplast, which produces phytoalexins and a different type of reactive oxygen species and changes the cell pH [29]. Melanin formation protects filamentous fungi against biotic stresses [30]. Therefore, to evaluate the role of ∆*FoATG12* and *OEFoATG12* in response to the stressors, we measured the effect of salt and osmotic stressors on the vegetative growth of mutants. When these mutants were grown on a PDA medium which contained 1.0 M Sorbitol and 0.7 M KCl, it was observed that the colony diameter of the WT and ∆*FoATG12* colonies was not significantly (*p* > 0.05) different from *OEFoATG12* (Figure 4B,D). Interestingly, WT and *OEFoATG12* showed a difference in color on both the media compared to ∆*FoATG12*. The WT and *OEFoATG12* strains exhibited an obvious change in color as compared with ∆*FoATG12* (Figure 4A,C). 

### 2.4. Plant Infection Assay

Potato tubers inoculated with the transgenic *F. oxysporum* strains were used to analyze the role of ∆*FoATG12* in determining the pathogenicity and virulence of *F. oxysporum*. Symptoms of the infection started to appear in potato tubers inoculated with the WT strains of *F. oxysporum* after 5 days. The disease symptoms started to appear after 7 days in tubers inoculated with the ∆*FoATG12*, indicating a slightly delayed disease development in the case of ∆*FoATG12* as compared to the WT strains, whereas the potato tubers inoculated with the *OEFoATG12* exhibited a similar pattern of disease development to the WT strains (Figure 5A). These findings strongly reflect the regulatory effect of the autophagy-related *ATG12* gene in the early stages of *F. oxysporum* infection in potato crops. An interesting finding revealed the non-virulence of the ∆*FoATG12* in the case of tomato leaves, as well as revealing that it was non-virulent on potato tubers. Significant progressive symptoms of wilt were exhibited by tomato seedlings infected with the *OEFoATG12* and WT strains (Figure 5B). After inoculation, the infected plants were unable to survive for 20 days, showing the disease severity and effective pathogenicity.

### 2.5. Expression of Pathogenicity and Vegetative Growth Related Genes

The qRT-PCR test was used to evaluate the genes associated with pathogenicity and vegetative growth in order to investigate whether *FoATG12* inhibits or stimulates the mechanism of gene expression involved in conidia production and pathogenicity in *F. oxysporum*. The analysis of the *FOXG_04522* gene expression, which plays a key role in conidia formation, revealed that the *FOXG_04522* gene did not showed expression in Δ*FoATG12*, as compared to WT mutants where it expressed itself. On the other hand, the pattern of expression observed in *OEFoATG12* mutants did not differ significantly from the WT strain (Figure 6C).

Several other important genes including *FOXG*_*NLP2* (NPPI2 domain-containing protein), *FOXG*_*SNF1* (sucrose non-fermentation protein kinase), *FOXG*_*VMK* (MAP kinase), and *FOXG*_*NLP1* (NPPI1 domain-containing protein), which are well known for their role in virulence against *F. oxysporum*, were also investigated. The results revealed that the expression level of *FOXG*_*VMK*, *FOXG*_*NLP1*, and *FOXG*_*SNF1* genes exhibited a significant decreasing trend in the Δ*FoATG12* as compared to the *OEFoATG12* and WT strains, whereas the expression of *FOXG_NLP2* also revealed a significant change in the Δ*FoATG12* (Figure 6A,B). The current findings strongly support the hypothesis that the expression of vegetative growth and virulence-related genes in the fungus *F. oxysporum* is significantly inhibited by the *ATG12* deletion mutant. 

## 3. Discussion

Autophagy is an intracellular recycling pathway in eukaryotic cells which is activated during a nutrient starvation condition. In starvation conditions, autophagy helps the cell survival, development, and differentiation [31]. The deletion mutant of *ATG8* inhibited the production of conidia in necrotrophic plant pathogenic fungi, *Fusarium graminearum*, as well as the effect development of aerial hyphae [32]. The null mutant of Pa*ATG1* and Pa*ATG8* in *Podospora anserine* reduced aerial hyphae formation and protoperithecia more than the wild-type strain [33]. In *V. dahlia*, the deletion mutant of Vd*ATG8* and Vd*ATG12* reduced conidial production and aerial hyphae formation, which reflects that autophagy plays a significant role in the supply of nutrients that are vital for the fungal growth and development under starvation. Recycling of organelles and cytosol endogenously is important for nutrient transport within hyphal filaments, the development of conidiophores, and aerial hyphae formation [34,35]. The deletion mutant of *FoATG8* showed a reduction in conidial formation and aerial hyphae formation in filamentous fungi [12,29,30,31,32,34,36]. In accordance with this idea, the present study revealed that the deletion of *ATG12* dramatically reduced conidial production and radial growth, suggesting that autophagy is necessary for the nutrient trafficking required for fungal growth.

Autophagy has emerged as a key mechanism for controlling the dynamics of virulence in pathogenic fungal strains. Evidence has revealed that the deletion mutants *MoATG12* and *MoATG8* failed to produce infection in rice [12]. In *Colletotrichum orbiculare*, the anthracnose fungus of cucumber lacking *CoATG8* was unable to infect cucumber cotyledons [5,6]. The deletion mutant of Vd*ATG8* and Vd*ATG12* exhibited a delay in the development of disease symptoms and reduced the overall pathogenicity in the *Arabidopsis* plant. These findings are in line with the previously reported results obtained from tomato infectious strains Dvd-T5. It was observed that, although it did not affect the overall virulence of the strain, the deletion of Vd*ATG8* significantly inhibited the colonization and spread of *V. dahlia* in tomato seedlings [37]. Similar findings were observed in deletion of *F. graminearum ATG8* [37]. *V. dahlia* is a soil-borne plant pathogen that causes infection via roots and penetration in the cuticles of plants [37]. These findings suggest that nutrient recycling by autophagy-dependent manors is necessary for the successful invasion of *V. dahliae* on the host plant. Our results show that, as compared to the WT and *OEFoATG12* mutants, the deletion of *FoATG12* resulted in delayed symptoms appearance in the *FoATG12* mutant. These findings strongly suggest that the autophagy-related *ATG12* gene may regulate the mechanism of infection and pathogenicity in the fungus *F. oxysporum* during the early stages of infection.

Plant pathogenic fungi always adapt a series of complex strategies for successful invasion on the host, which includes the mechanism of plant defense, adapting the intracellular environment of the host and defeating adverse environmental changes [38,39]. In various phytopathogens, some genes and factors, such as bZIP TF MoAP1, YAP1 homologue, and oxalic acid, are involved in the mediation of oxidative burst suppression and oxidative stress tolerance of the host pathogen [7,38,40,41]. The vegetative growth of DVdpf mutants, IMDVdpf, WT, and Com strains in oxidative stress conditions, such as H_2_O_2_ and Sorbitol, had a similar effect. The Vdpf is unnecessary for the detoxification of ROS and osmo regulation [42]. In present study, we found that the oxidative stressors, such as KCl and Sorbitol, showed similar results in the vegetative growth of ∆*FoATG12*, *OEFoATG12*, and WT. Interestingly, the difference of colony morphology was observed in ∆*FoATG12* as compared to WT and *OEFoATG12*.

The *FoATG12* deletion mutant significantly inhibited the expression of pathogenicity and virulence-related genes in the potato tuber infection assay. VdSNF1 and VdPKAC1 genes are well-recognized causal factors of virulence in tomato and eggplants. Studies have proven that the VMK1 gene shows effective virulence in potato and tomato plants, whereas VdNLP2 and VdNLP1 genes are key virulence factors in tobacco, Arabidopsis, and cotton plants [13,14,15,27,37,42]. The gene factors which determine the virulence in *F. oxysporum*, including *FOXG_NLP1* (NPPI1 domain-containing protein), *FOXG_VMK* (MAP kinase), *FOXG_SNF1* (sucrose non-fermentation protein kinase), and *FOXG_NLP2* (NPPI2 domain-containing protein), were investigated in the present study. Our findings strongly recommend that the autophagy-related gene *ATG12* inhibits the expression level of vegetative growth and that pathogenicity related genes may regulate the mechanism of infection in *F. oxysporum*.

## 4. Materials and Methods

### 4.1. Isolation of the Fungal Strain and the Culture Conditions

*F. oxysporum* strains were isolated from the infected potato tubers, and their identity was confirmed by plant infection assay before being used as a wild-type (WT) strain in the present study. The isolated strains were sensitive to the HygB at concentrations > 30 mg/mL and were subsequently used as WT strains in the experiment. The conidial suspension of the isolated strains was made followed by the addition of 50% glycerol and then stored at −80 °C. Fungal mycelium and conidia were grown in a liquid potato dextrose agar (PDA) medium. The conidia were reactive at 25 °C on the fresh PDA medium for further use. The genetic transformation of *F. oxysporum* was carried out by using specific strains (GV3101) of *A. tumefaciens*, which were cultured on liquid broth (LB) media. 

### 4.2. Fungal Transformation

The process of the fungal transformation was executed following the previously described ATMT method with a few modifications [43]. *A. tumefaciens* GV3101 strains equipped with the vector pPK2 were grown in PDA-amended media at 28 °C. The culture of bacterial cells was mixed with the conidial suspension in equal concentrations (10^7^/mL) when the optical density values reached 6600 nn (OD_660_) at 0.5, they were further diluted in the induction medium containing acetosyringone (AS) (200 mM) to a value of OD_600_ at 0.15. It was again cultured for 6 h on an orbital shaker (200 rpm) at 28 °C. About 250 mM of solution from this mixture was kept as a co-cultivation medium for 48 h on 80 nm nitrocellulose filers with a 0.45 mm pore size (Whatman, Japan). The filters were transferred to a HygB (50 mg/mL)- and cefotoxime (500 mg/mL)-amended selective medium to defeat the *A. tumefaciens* cells. After 7 days, the randomly selected transformants were again cultured on HygB (50 mg/mL)-enriched PDA media. 

### 4.3. Generation of FoATG12 Deletion Mutants

Fusion PCR was used for the generation of the *FoATG12* deletion mutant (Szewczyk, Lam) [44,45]. The mutants were generated by using a HygB resistance cassette with the replacement of a 1040 bp open reading frame (ORF). Three primer pairs, including P1/P2, P3/P4, and P5/P6, were used to amplify the downstream 1000 bp fragment of *FoATG12* along with the HygB resistance cassette obtained from the vector psilent-1. The three fragments were fused with U-Hph-D followed by digestion with the vector pPK2-ligated restriction enzymes AsiSI and SbfI (Appendix A). The final transmission of the pPK2-U-Hph-D recombinant plasmid into WT was carried out following the previously described method [43]. The transformants were screened by using PDA media augmented with cefo (50 µM) and of HygB (50 mg/L). The deletion mutants were identified, and the transformation process was confirmed by using F-hph/R-hph PCR screening. 

### 4.4. Overexpression (OE) of the Mutant FoATG12 Strains

Overexpression (*OE*) of the mutant *FoATG12* was achieved by cDNA of *F. oxysporum*, which completely encompassed the ORF of *F. oxysporum*. The primer *ATG12F/R* was used for amplification of the total RNA, which corresponded to the initial 7 ORF codons and additional cytosine at restriction sites NotI, along with the *ATG12*-R reverse complement at the SbfI restriction site with additional cytosine. The resulting band was cloned to the p8GWN vector, which was used as a template for PCR amplification. The GFP *ATG12*-F primer corresponded to the 1st 8 codons of GFP, with ORF at restriction sites AscI and SbfI (Appendix A). The GFP-*FoATG12*/F303 plasmid was obtained by cloning the DNA-amplified fragment into F303 vector. The primer pair M13F and M13R was used for PCR amplification to obtain a 3.4 kb fragment having *FoATG12 GFP* fusion controlled by a *trpC* promoter as well as terminator (Appendix A).

### 4.5. Evaluation of Fungal Germination, Conidia Formation, and Radial Growth

PDA media containing 50 mg/mL HygB were used to investigate the radial growth and determine the conidial growth in the *F. oxysporum* strains. Conidial growth was measured by harvesting conidia from a 10 days older fungal culture. The harvested two-layered conidia were filtered and resuspended in sterile water in a 1 × 10^7^ conidia/mL concentration. The conidial suspension (5 µL) of ∆*FoATG12*, *OEFoATG12*, and WT was inoculated, and the plates were incubated at 25 °C followed by daily measurements of the colony diameters. The conidial germination was observed at 7, 12, 21, 28, 36, and 48 h, respectively, by inoculating conidia (10^2^) in the PDB medium (1 mL) with continuous shaking at 150 rpm, and the germination rate was calculated after 12 h using a blood-counting chamber (Di Pietro) [46]. The experimental data were statistically analyzed by using SPSS software 15.0 for Windows^®^ (LEAD Technologies, Inc., Charlotte, NC, USA). The Duncan’s post hoc test (*p* < 0.05) was applied to reveal the differences among the investigated strains. 

### 4.6. Evaluation of the Fungal Sensitivity against Stress

To evaluate the sensitivity of strains against salt and osmotic stress, 5 µL droplets of the conidial suspension of WT and other mutants were pipetted on plates of PDA which contained 1.0 M Sorbitol and 0.7 M KCl. The diameter of the colonies was measured after 7 days of inoculation. All experiments were repeated three times.

### 4.7. Analysis of Gene Expression

RNA extraction was carried out from *F. oxysporum* WT strains as well as ∆*FoATG12* and *OEFoATG12* mutants, which was followed by qRT PCR. RNA extraction was carried out using Magnet (mini kit-Hi pure RNA) with the protocol script using a mixture of oligo (dT) and Rt Enzyme (Takara) enzymes with RNA. Bio-Rad PTC0200 Peltier Thermal Cycler was used to run the RT PCR (30 cycles) (Bio-Rad, Hercules, CA, USA) after diluting the cDNA to 100 ng/1 L. EF1α gene amplification was done for the sake of internal control, and the resultant gray scale was used to reveal the expression values. We used a 1st template standard of cDNAs along with TB SYBER supermiz (Takara) for the qRT-PCR analysis. All the reactions were repeated three times while using EF1α as an endogenous control. The PCR protocol followed for running qRT-PCR started with an initial denaturation for 2 min at 95 °C, which was followed by 40 cycles (95 °C) for 10 s. An annealing temperature of 60 °C was attained for *FOXG_ NLP1* (NPPI1 domain-containing protein), *FOXG_SNF1* (sucrose non-fermentation protein kinase), *FOXG_VMK* (MAP kinase), and *FOXG_NLP2* (NPPI2 domain-containing protein) for 30 s, whereas the temperature was 63.5 °C for *FOXG_04522*, respectively. CFX Manager^TM^ was used to analyze the data. The comparative Ct method (2^−∆∆Ct^) was used to determine the normalized expression level among the WT and the mutants as ∆∆Ct = (Ct_gene_ − Ct_18srRNA_)_mutant_ − (Ct_gene_ − Ct_18srRNA_)_WT_. The specific gene primer pairs were designed to be used for the RT-PCR, as listed in Appendix A. All the experiments performed with the biological replicates were repeated three times.

### 4.8. Microscopic Analysis

The detailed microscopic analysis was done with the cell aliquots embedded in 1% agarose solution blocks, which were studied using appropriate filter sets under the microscope (M2 Dual Cam) using UV light with a 340–380 nm wavelength range. A photometric digital camera (Evolve EM512) equipped with Axiovision software (version 4.8) was used to record the images, which were further processed using Adobe Photoshop (CS-3). 

### 4.9. Pathogenicity Test

Uniform-sized, healthy potatoes (100–120 g) cv. Desiree were used for the inoculation assay. Contamination and excess soil were removed by thorough washing of the tubers followed by a surface sterilization with 0.5% sodium hypochlorite solution for 10 min. The treated tubers were then rinsed in three changes with sterile distilled water. The tubers were then cut into small slices and were air dried. The mutant strains of *FoATG12*, *OEFoATG12*, ∆*FoATG12*, and WT were incubated for 14 days on TRA medium in darkness. The sporangia for each strain were collected and then washed using pea broth. A 120-mL sporangial solution with a concentration of 1 × 10^4^/mL was inoculated on the dried potato slices for each of the *FoATG12*, ∆*FoATG12*, and WT strains. Five potato slices for each treatment were placed in three replicates on a moist filter paper placed in a dish and were incubated for 5 days under conditions at 27 °C. After germination, the disease severity of each fungal strain was recorded.

Finally, 2-week-old tomato cv. Terminator seedlings were inoculated with the *F. oxysporum* for the tomato inoculation assay following the method given by Di Pietro et al. [46]. The seedling roots were immersed in a fungal suspension with a concentration of 5 × 10^6^ spores/mL for 30 min and were sown in a vermiculated growth chamber. Overall, 10 plants for each of the treatments were used. The disease severity was recorded on a daily basis [46]. The experimental data was statistically analyzed by using SPSS software 15.0 for Windows^®^ (LEAD Technologies, Inc.). Duncan’s post hoc test (*p* < 0.05) was used to analyze the values with each experiment repeated three times.

## Figures and Tables

**Figure 1 ijms-22-04932-f001:**
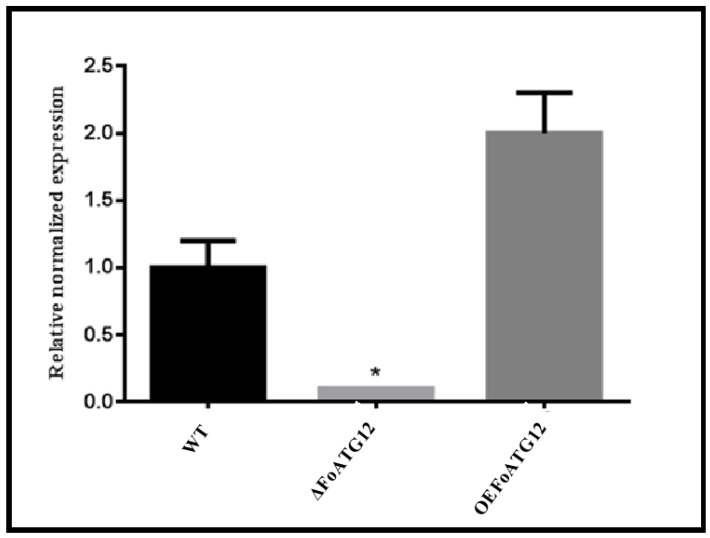
Verification of mutants. Quantitative real-time reverse-transcription polymerase chain reaction (qRT-PCR) analysis of wild-type (WT), ∆*FoATG12*, and overexpression (*OEFoATG12*) strains. Three biological replicates were used for this study. Duncan post hoc test (*, *p* < 0.05) was performed for the statistical analysis, whereas the error bars depict the standard deviation in the values.

**Figure 2 ijms-22-04932-f002:**
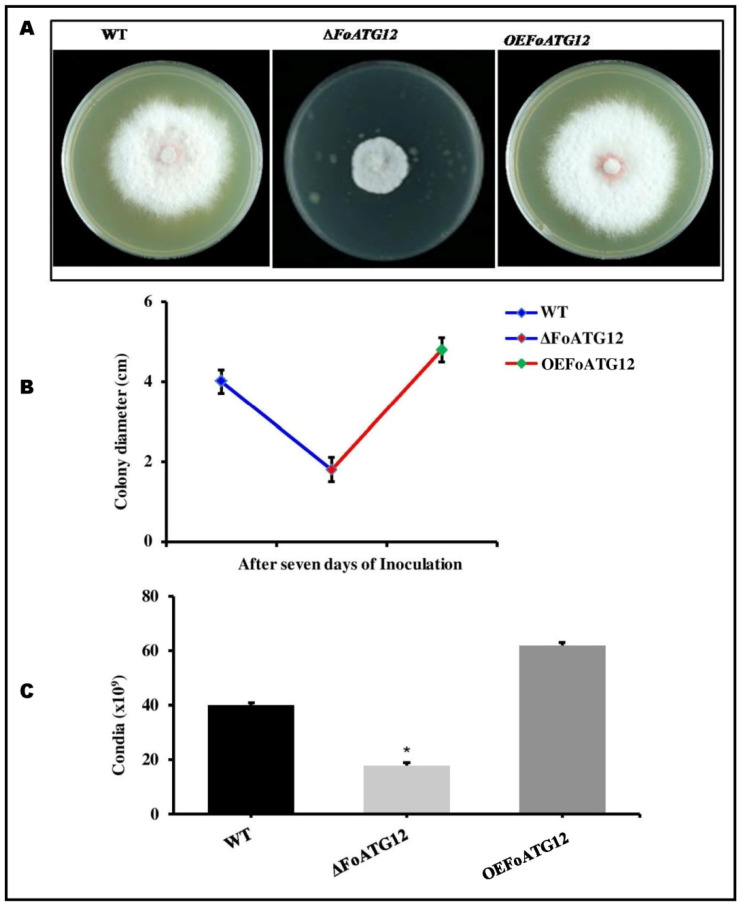
*FoATG12* mutants showed a reduced conidiation and hyphal growth. (**A**) The investigated fungal strains after 7 days of inoculation. (**B**) Growth curves of the measured colony diameters (7 days) for the investigated fungal strains. Fresh microconidia (10^3^) were cultured on a PDA medium and incubated for 7 days at 28 °C. (**C**) The recovered microconidia obtained after 7 days from the PDA medium. Duncan post hoc test (*, *p* < 0.05) was used with the error bars showing the standard deviation.

**Figure 3 ijms-22-04932-f003:**
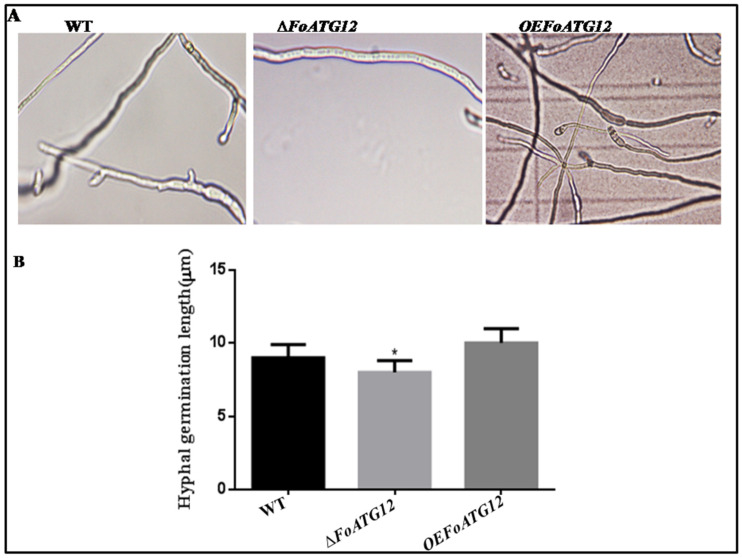
*FOATG12* mutants showed reduced length and hyphal growth. (**A**) Showing the density of the germinated fungal hyphae. (**B**) Depicting the length of the fungal hyphal germination. Duncan post hoc test (*, *p* < 0.05) was used with the error bars showing the standard deviation.

**Figure 4 ijms-22-04932-f004:**
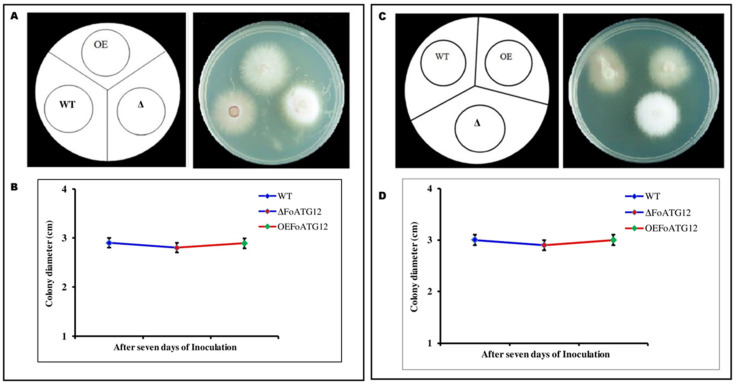
Representative phenotypes of wild-type (WT), deletion mutant (∆*FoATG12*), and overexpression (*OEFoATG12*). (**A**) Represents phenotype comparison between WT, ∆*FoATG12*, and *OEFoATG12* on 1.0 M Sorbitol. (**B**) Growth curves of the measured colony diameters (7 days) for the investigated fungal strains. Fresh microconidia (10^3^) were cultured on a PDA medium and incubated for 7 days at 28 °C. (**C**) Represents phenotype comparison between WT, ∆*FoATG12*, and *OEFoATG12* on 0.7 M KCl. (**D**) Growth curves of the measured colony diameters (7 days) for the investigated fungal strains. Fresh microconidia (10^3^) were cultured on a PDA medium and incubated for 7 days at 28 °C. Colony diameter represents a non-significant (*p* > 0.05) difference after 7 days of inoculation, whereas the ∆*FoATG12* showed a striking difference in color.

**Figure 5 ijms-22-04932-f005:**
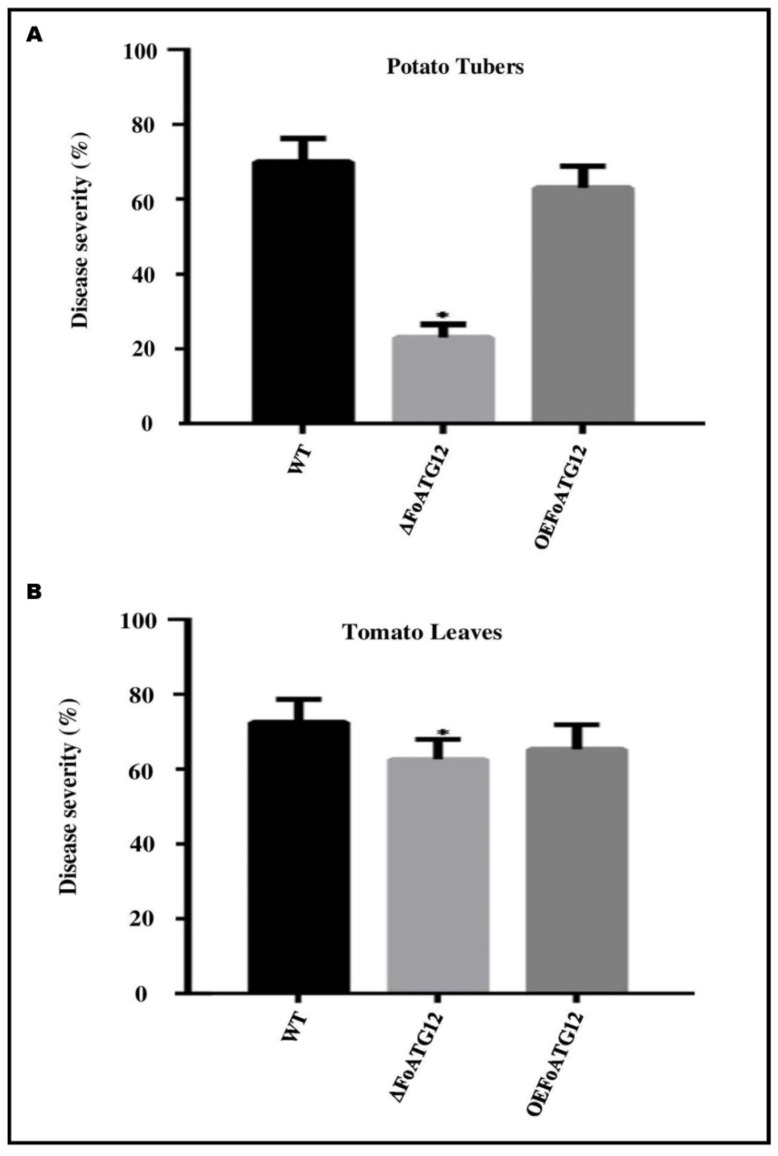
*FoATG12* gene significantly contributes to virulence of *F. oxysporum*. Figure represents the disease progress in (**A**) potato tubers and (**B**) tomato leaves. Potato tubers and tomato leaves were inoculated with a freshly harvested microconidial suspension (5 × 10^6^) of *FOATG12 OEFoATG12*, *WT*, and Δ*FoATG12* strains. The disease symptoms were observed after 20 days of inoculation. The experiments were repeated three times with similar results and statistically investigated by Duncan post hoc test (*, *p* < 0.05) with the error bars showing the standard deviation.

**Figure 6 ijms-22-04932-f006:**
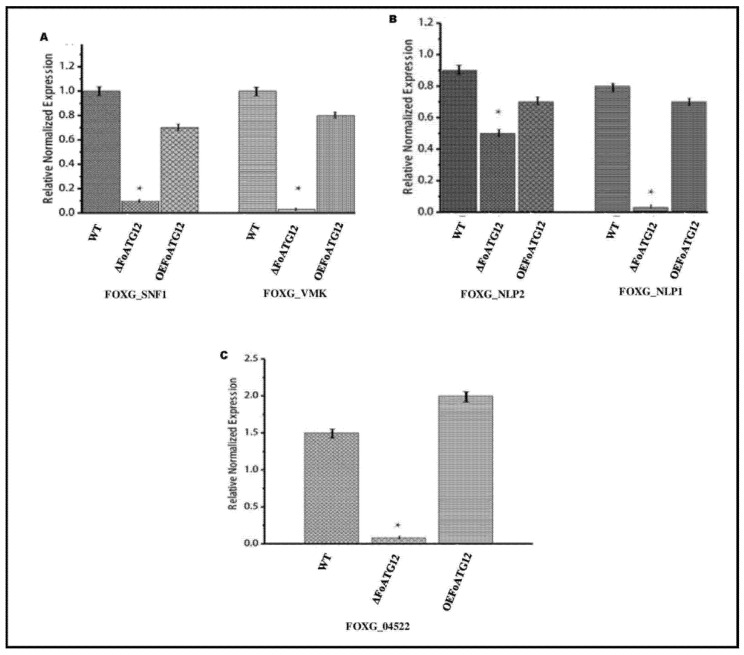
Figure showing the normalized expression of pathogenicity and vegetative growth related genes. (**A**) Represents the expression of pathogenicity related genes *FOXG_SNF1* and *FOXG_VMK*, (**B**) represents the expression of pathogenicity related genes *FOXG_NLP2* and *FOXG_NLP1* and (**C**) represents normalized expression of vegetative growth-related genes *FOXG_04522*. The relative values of transcription abundance were calculated for each gene. The normalized relative quantity of the transcripts is illustrated by the *y*-axis. Standard deviation is indicated by the error bars in the graph, whereas Duncan’s post hoc test (*, *p* < 0.05) was applied for statistical analysis.

## Data Availability

Not applicable.

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
