# Peer review of "Functional Analysis of Autophagy-Related Gene ATG12 in Potato Dry Rot Fungus Fusarium oxysporum"

_ijms, 2021, doi:10.3390/ijms22094932_

Round 1
Reviewer 1 Report
The manuscript of Khalid et al. describes the role of FoATG12 gene in regulation of growth and virulence of Fusarium oxysporum. The authors obtained mutated trains of fungus with either deletion or over expression of the gene of interest and compered their ability to growth under optimal conditions, stress resistance and virulence potential. The manuscript contains results which mostly stay in line with numerous previous reports on the role of autophagy genes in functioning of plant fungal pathogens. Nevertheless, some issues must be corrected and clarified.
- The manuscript requires major linguistic corrections. The meaning of some sentences and phrases is impossible to be perceive precisely.
- Please unify the writing of the names of your fungus lines in all manuscript giving they full names, not “Ko” or “OE”. Besides, I guess that on Fig. 1 authors mean OEATG12 not 22.
- Please unify and precise all figures captures. What present panel A and B on Fig. 4? Please measure the diameters of colonies (Fig. 4) as it was performed on Fig. 2.
- Fig. 5 – Please explain why the survival rate of potato tubers is lower after infection with deletion mutant? According to your results this line should posses lower infection potential comparing to wild type strain.
Please write clearly how the measurement of survival rate of plants was performed (what and how was measured). Please provide appropriate photographs to illustrate infection symptoms on plants. Some other parameters of plants response to infection caused by different lines of fungus should be provided, e.g. plant marker genes expression, ROS accumulation, dead cells staining, etc.
- Please provide the full names of genes which expression was monitored and presented at Fig. 6. Why it is stated in the text that expression of FOX_NLP2 is unchanged in deletion mutant? On the Fig. 6B it is clearly two times lower than in wt strain and the difference is statistically relevant.
- I suggest to organize and clearly express Authors postulates in the discussion. The part of discussion concerning H2O2 effect on fungus growth is completely incomprehensible – the H2O2 effect on fungus growth was not tested and presented in the manuscript.
- Please apply Italic if necessary.
Author Response
Manuscript (ijms-1164017)
Response to Reviewers #1
Dear Sir,
Thank you for giving us the opportunity to submit a revised draft of the manuscript “Functional Analysis of Autophagy Related Gene ATG12 in Potato Dry Rot Fungus Fusarium oxysporum
” for publication in the Journal of IJMS. We appreciate the time and effort that you and the reviewers dedicated to providing feedback on our manuscript and are grateful for the insightful comments on and valuable improvements to our paper. We have incorporated most of the suggestions made by the reviewers. Those changes are highlighted within the manuscript. Please see below, in red, for a point-by-point response to the reviewers’ comments and concerns. All page numbers refer to the revised manuscript file with tracked changes.
Reviewers' Comments to the Authors:
Response to Reviewer #1:
We would like to thank the reviewer for careful and thorough reading of this manuscript and for
the thoughtful comments and constructive suggestions, which help to improve the quality of this
Manuscript. Our response follows (the reviewer’s comments are in italics).
General Comments.
The manuscript of Khalid et al. describes the role of FoATG12 gene in regulation of growth and virulence of Fusarium oxysporum. The authors obtained mutated trains of fungus with either deletion or over expression of the gene of interest and compered their ability to growth under optimal conditions, stress resistance and virulence potential. The manuscript contains results which mostly stay in line with numerous previous reports on the role of autophagy genes in functioning of plant fungal pathogens. Nevertheless, some issues must be corrected and clarified.
Reply:
We appreciate the positive feedback from the reviewer.
With regards to extensively revision and correction of manuscript as we noted in our response to Reviewer #1. According to kind suggestions of reviewer we have reviewed carefully the entire manuscript and have removed redundancies, as shown in the revised manuscript.
Major Comments:
1) The manuscript requires major linguistic corrections. The meaning of some sentences and phrases is impossible to be perceive precisely.
Reply:
As suggested by reviewer whole manuscript has been revised carefully and correction has been made as suggested.
2) Please unify the writing of the names of your fungus lines in all manuscript giving they full names, not “Ko” or “OE”. Besides, I guess that on Fig. 1 authors mean OEATG12 not 22.
Reply:
Corrections has been made as suggested.
3) Please unify and precise all figures captures. What present panel A and B on Fig. 4? Please measure the diameters of colonies (Fig. 4) as it was performed on Fig. 2.
Reply
According to kind suggestion of reviewer all figures captions has been unified and précised. Corrections has been made.
4) Fig. 5 – Please explain why the survival rate of potato tubers is lower after infection with deletion mutant? According to your results this line should posses lower infection potential comparing to wild type strain.
Reply:
Now line 158, It was typing mistake. Furthermore, corrections has been made according to kind suggestions of reviewer.
5) Please write clearly how the measurement of survival rate of plants was performed (what and how was measured). Please provide appropriate photographs to illustrate infection symptoms on plants. Some other parameters of plants response to infection caused by different lines of fungus should be provided, e.g. plant marker genes expression, ROS accumulation, dead cells staining, etc
Reply:
Now in line 360, it is mentioned that how survival rate of plants was measured. Furthermore corrections has been made according to suggestions.
6) Please provide the full names of genes which expression was monitored and presented at Fig. 6. Why it is stated in the text that expression of FOX_NLP2 is unchanged in deletion mutant? On the Fig. 6B it is clearly two times lower than in wt strain and the difference is statistically relevant.
Reply:
Now line 188, it was typing mistake now correction has been made according to kind suggestions.
7) I suggest to organize and clearly express Authors postulates in the discussion. The part of discussion concerning H2O2 effect on fungus growth is completely incomprehensible – the H2O2 effect on fungus growth was not tested and presented in the manuscrip.
Reply:
Now line 243-244, Text has been made according to suggestions.
8) Please apply Italic if necessary.
Reply:
Whole manuscript has been revised and italic has been applied where necessary according to kind suggestions.
9) What the cultivar of Tomato used in the inoculation assay? How the data were analyzed?
Reply
Now line 355 & 363, correction has been made.

Reviewer 2 Report
Font sizes mismatch in 3rd paragraph of Introduction;
2.1, in the sentence, “over expression of the FoATG12 which was further validated by the PCA results in which the expected overexpression was shown by the transformants”, “PCA”? PCR?
What is the * in figure 1, 2 and 3?
The figure legends should be self-explanatory. All figures’ legends in the manuscript are lack of enough info.
2.2, when reporting “reduced significantly” or “inhibits hyphal growth compare to WT and OE”, or “not significantly”, add p value for these statements.
More detailed info about the origin of the F. oxysporum isolates were required; The procedures of how the strains were isolated were also missing in 3.1. The isolated pathogens need molecular confirmation as F. oxysporum. The origin of A. tumefaciens GV3101?
How data were analyzed in 3.6?
Which potato cultivar were used in 2.9 (Should be 3.9)?
What the cultivar of Tomato used in the inoculation assay? How the data were analyzed?
Author Response
Manuscript ID: ijms-1164017
Dear Sir,
Thank you for giving me the opportunity to submit a revised draft of my manuscript titled “Functional Analysis of Autophagy Related Gene ATG12 in Potato Dry Rot Fungus Fusarium oxysporum”. We appreciate the time and effort that you and the reviewers have dedicated to providing your valuable feedback on my manuscript. We are grateful to the reviewers for their insightful comments on my paper. We have been able to incorporate changes to reflect most of the suggestions provided by the reviewers. We have highlighted the changes within the manuscript. Here is a point-by-point response to the reviewers’ comments and concerns.
Comments from Reviewer #2
- Comment 1: Font sizes mismatch in 3rdparagraph of Introduction;
Response:
Changes has been made as suggested. Thank you for pointing this out.
- Comment 2: 2.1, in the sentence, “over expression of the FoATG12 which was further validated by the PCA results in which the expected overexpression was shown by the transformants”, “PCA”? PCR?
Response:
Now in line #92, changes has been made accordingly.
- Comment 3: What is the * in figure 1, 2 and 3?
Response:
Represents significant difference.
- Comment 4: The figure legends should be self-explanatory. All figures’ legends in the manuscript are lack of enough info.
Response:
Thank you for this suggestion. According to kind suggestions figure legends gas been revised and changes has been made accordingly.
- Comment 5: 2.2, when reporting “reduced significantly” or “inhibits hyphal growth compare to WT and OE”, or “not significantly”, add p value for these statements.
Response
Now in line 113, 114, Changes has been made as suggested.
- Comment 6: More detailed info about the origin of the F. oxysporum isolates were required;The procedures of how the strains were isolated were also missing in 3.1. The isolated pathogens need molecular confirmation as F. oxysporum. The origin of A. tumefaciens GV3101?
Response:
Now in line # 259, Changes has been made according to kind suggestions of reviewer.
- Comment 7: How data were analyzed in 3.6?
Response:
Now in line 318, Changes has been made as suggested.
- Comment 8: Which potato cultivar were used in 2.9 (Should be 3.9)?
Response
Uniform sized, healthy potatoes (100-120g) cv. Desiree were used for the inoculation assay. Now in line #346 changes has been made.
- Comment 9: which tomato cultivar was used in the inoculation assay? How the data was analyzed?
Response
Two weeks old tomato cv. Terminator seedlings were inoculated with the F. oxysporum for the tomato inoculation assay. Now in line # 364, Changes has been made.

Round 2
Reviewer 1 Report
Dear Authors,
thank you for your work and improvement of the manuscript.
Please take a look at some minor issues:
- The over expression line name writing – we can find OEATG12 and OEFoATG12 in the text and on Figures.
- The y axis name “Survival” at Fig. 5 is misleading. It is rather the size of fungus colony in a case of potato tubers and the % of wilted leaves for tomato plants, as stated in the Methods chapter. Thus I recommend dividing this graph into separate ones with adequate y axis names.
- Full names of genes ( FOXG_SNF1, _VMK, _NLP1, _NLP2) should be given when mentioned for the first time.
Author Response
Response to Reviewers #1
Dear Sir,
Thank you for giving us the opportunity to submit a revised draft of the manuscript “Functional Analysis of Autophagy Related Gene ATG12 in Potato Dry Rot Fungus Fusarium oxysporum
” for publication in the Journal of IJMS. We appreciate the time and effort that you and the reviewers dedicated to providing feedback on our manuscript and are grateful for the insightful comments on and valuable improvements to our paper. We have incorporated most of the suggestions made by the reviewers. Those changes are highlighted within the manuscript. Please see below, in red, for a point-by-point response to the reviewers’ comments and concerns. All page numbers refer to the revised manuscript file with tracked changes.
Reviewers' Comments to the Authors:
Minor Comments:
1) The over expression line name writing – we can find OEATG12 and OEFoATG12 in the text and on Figures.
Reply:
As suggested by reviewer line name has been revised carefully and correction has been made as suggested.
2) The y axis name “Survival” at Fig. 5 is misleading. It is rather the size of fungus colony in a case of potato tubers and the % of wilted leaves for tomato plants, as stated in the Methods chapter. Thus I recommend dividing this graph into separate ones with adequate y axis names.
Reply:
Corrections has been made as suggested.
3) Full names of genes (FOXG_SNF1, _VMK, _NLP1, _NLP2) should be given when mentioned for the first time.
Reply
According to kind suggestion, corrections has been made
